# Targets (Metabolic Mediators) of Therapeutic Importance in Pancreatic Ductal Adenocarcinoma

**DOI:** 10.3390/ijms21228502

**Published:** 2020-11-12

**Authors:** Vikrant Rai, Swati Agrawal

**Affiliations:** 1Department of Translational Research, College of Osteopathic Medicine of the Pacific, Western University of Health Sciences, Pomona, CA 91766, USA; 2Department of Surgery, Creighton University School of Medicine, Omaha, NE 68178, USA; swatiagrawal@creighton.edu

**Keywords:** PDAC, metabolomics, metabolic mediators, therapeutic targets

## Abstract

Pancreatic ductal adenocarcinoma (PDAC), an extremely aggressive invasive cancer, is the fourth most common cause of cancer-related death in the United States. The higher mortality in PDAC is often attributed to the inability to detect it until it has reached advanced stages. The major challenge in tackling PDAC is due to its elusive pathology, minimal effectiveness, and resistance to existing therapeutics. The aggressiveness of PDAC is due to the capacity of tumor cells to alter their metabolism, utilize the diverse available fuel sources to adapt and grow in a hypoxic and harsh environment. Therapeutic resistance is due to the presence of thick stroma with poor angiogenesis, thus making drug delivery to tumor cells difficult. Investigating the metabolic mediators and enzymes involved in metabolic reprogramming may lead to the identification of novel therapeutic targets. The metabolic mediators of glucose, glutamine, lipids, nucleotides, amino acids and mitochondrial metabolism have emerged as novel therapeutic targets. Additionally, the role of autophagy, macropinocytosis, lysosomal transport, recycling, amino acid transport, lipid transport, and the role of reactive oxygen species has also been discussed. The role of various pro-inflammatory cytokines and immune cells in the pathogenesis of PDAC and the metabolites involved in the signaling pathways as therapeutic targets have been previously discussed. This review focuses on the therapeutic potential of metabolic mediators in PDAC along with stemness due to metabolic alterations and their therapeutic importance.

## 1. Introduction

Pancreatic ductal adenocarcinoma (PDAC) is the fourth most leading cause of cancer-related death in the United States. The 5-year disease-free survival rate for PDAC patients after therapy is less than 8–10% [1,2,3]. The inability to detect the disease early and refractoriness of PDAC to existing treatments are attributes to the grimness of PDAC [4]. PDAC has been projected to be the second most common cause of cancer-related death by 2030 [5]. Current therapies for PDAC including surgical intervention combined with neoadjuvant or adjuvant chemotherapy, immunotherapy, palliative care (surgical intervention) including relief from obstructive jaundice, duodenal or gastric outlet obstruction, and pain have provided better results and increased survival. However, the targeting of cancer-associated molecular pathways has not given satisfactory results and has failed to provide meaningful clinical benefits [6,7,8]. This may be due to the molecular heterogeneity between the unique advanced PDAC genomic and transcriptomic subtypes between the individual cases [9]. The heterogeneity of the PDAC might be due to carcinoma-associated fibroblasts (CAFs) [10], which make up the bulk of the tumor stroma and are associated with increased growth, suppress the immune response, and enhance metastatic dissemination. CAFs promotes tumorigenic features by deposition and initiating the remodeling of the extracellular matrix, secreting cytokines, extensive reciprocal interactions with cancer cells and crosstalk with infiltrating leukocytes [10,11]. Therapeutic resistance is another potential problem when dealing with PDAC patients. Surgical resection of the tumor tissue as a curative treatment is available for the tumors contained within the pancreas and not for locally advanced (American Joint Commission on Cancer (AJCC) stage III) or metastatic PDAC (AJCC stage IV). Surgical resection of PDAC can also be associated with recurrent PDAC and poor quality of life. A precise identification of the patients who would benefit from the surgical resection is of utmost importance. Radiologically, the PDAC for surgical resection can be divided into three categories—resectable, borderline resectable, and unresectable [3]—and both computed tomography and magnetic resonance imaging aid in the surgical staging of PDAC. The definition and criteria to classify PDAC in these categories have been reviewed [3]. An accurate identification of borderline resectable PDAC is important and should be clearly distinguished from both resectable and locally advanced primary tumors because these patients are at higher risk for perioperative complications. The tumors which abut the superior mesenteric artery and the common hepatic artery over a short segment, or occlude the superior mesenteric vein-portal vein confluence with suitable vein above and below with a possibility of venous reconstruction are defined as borderline resectable PDAC. Additionally, the patients with extensive medical comorbidities requiring prolonged evaluation not needing immediate major abdominal surgery or presenting with indeterminate or questionable metastatic disease also falls under the borderline resectable category [12]. Due to the complexity of surgical intervention, higher peri- and post-operative complications, patients in the borderline resectable category should be treated using multidisciplinary approach involving neoadjuvant systemic chemotherapy and/or chemoradiation and surgery. The inadequacy of the medical treatment and complications associated with surgical intervention suggest an urgent need to investigate novel therapeutic targets for a better outcome.

A better understanding of the pathophysiology and the changes occurring at the molecular level within the tumor is a must. At present, the focus is on immune modulation (immunotherapy) and using chemotherapy or radiation therapy to either inhibit the growth or destroy the tumorous tissue [4]. The reprogramming of the metabolic pathways of carbohydrates, proteins, lipids, and nucleic acids fuel the growing energy needs of the tumor cells for their enduring survival and uncontrolled proliferation. Increased generation of ATP by switching to aerobic glycolysis (Warburg effect) for growing energy needs, biosynthesis of macromolecules for tumor mass synthesis, intermediates for anaplerotic reactions, and maintenance of redox status within the tumor microenvironment facilitate tumor growth [4,13,14]. The Warburg effect or metabolic alterations as seen in pancreatic cancer are characterized by enhanced glucose uptake, flux into glycolysis, and decreased glucose carbon flux into the tricarboxylic acid cycle (TCA) cycle, even in the presence of oxygen. The metabolic alterations of increased aerobic glycolysis and decreased oxidative metabolism result in the production of lactic acid, which creates an acidic tumor environment. Metabolic alterations are the hallmark of the tumor and are facilitated by the hypoxic and acidic tumor microenvironment as well as K-ras mutations. These have emerged as novel targets for therapeutic intervention [15,16,17]. In PDAC, deregulation of the glucose and glutamine metabolism pathways have been suggested as the cause of tumor growth and metastasis [4,18,19]. Further, different subtypes of PDAC show distinct metabolite profiles in association with glycolysis, lipogenesis, redox pathways, and correlation with enriched sensitivity to inhibitors targeting glycolysis, glutaminolysis, lipogenesis, and redox balance [20]. This is indicative of the need for a deep understanding of PDAC cell survival in the tumorous environment by metabolic reprogramming supporting tumor growth and metastasis to develop better therapies to circumvent the therapeutic insufficiency and resistance. This review focuses on summarizing the recent findings and the potential metabolic intermediates of therapeutic importance in the treatment of PDAC.

## 2. Tumor Microenvironment

Tumor microenvironment (TME) is characterized by an extremely dense fibrous stroma consisting of proliferating myofibroblasts (pancreatic stellate cells), deposition of type I collagen, hyaluronic acid, extracellular matrix (ECM) components, and various inflammatory cells, including macrophages, mast cells, lymphocytes, and plasma cells. Dense fibrous stroma not only contribute to increased microscopic arteriolar-venular shunts, low microvascular density, leaky vasculature, restricted perfusion, and reduced blood flow leading to intratumoral hypoxia, harsh, and nutrient-deprived conditions but also to tumor survival and growth by increased production of connective-tissue growth factors, cytokines, and deposition of ECM. Increased survival of the tumor cells is also facilitated by impaired drug delivery due to high interstitial fluid pressure in TME. Additionally, restriction of immune surveillance and the presence of chronic inflammation in the tumor microenvironment supports tumor growth [4,21]. Moreover, the altered glucose, aminoacids, and lipid metabolism and metabolic crosstalk between tumor cells and other nonmalignant cells within TME contribute to the increased tumor growth and poor survival of PDAC patients. Various components of TME could be an alternate source of energy such as cleaved collagen fragments and collagen-derived proline entering the TCA cycle through micropinocytosis—that is, pancreatic stellate cell (PSC)-secreted leukemia inhibitory factor and PSC-secreted alanine entering the TCA cycle as an alternative source of ATP in the harsh tumor environment [22]. Further, exosomes released by carcinoma-associated fibroblasts, a component of TME, contain amino acids, lipids, and TCA cycle intermediates also act as an alternative source of energy under nutrient-deprived conditions [23]. Similarly, the role of tumor-associated macrophages (TAMs) with elevated glycolytic signature and promoting pancreatic cancer vascularization and metastasis, increased procancer M2-like polarization of TAMs with lactate in TME, increased tumor progression mediated by the interactions of tumor cells and adipocytes in TME, and interaction of adipocytes with PSCs and tumor-associated neutrophils has been discussed in the literature [24]. Collectively, TME components play a crucial role in tumorigenesis through the tumor and nontumor cell interaction, escape from immune surveillance, and metabolic alterations in a harsh, hypoxic, and nutrient-deprived environment.

## 3. Metabolic Pathways

### 3.1. Glucose Metabolism

Glycolysis or glycolytic flux, a central carbon metabolism pathway in cells, provides energy and biomass for cell growth, division, and proliferation. Dramatically increased glycolytic flux is the hallmark of cancers even in the presence of normal mitochondrial function. Glycolytic flux is controlled by glucose transporters, rate-limiting enzymes, and the intermediates of glycolysis which regulate redox homeostasis, glycosylation, and biosynthesis. Further, the pentose phosphate pathway (PPP), hexosamine biosynthesis pathway (HBP), serine biosynthesis, and tricarboxylic acid cycle (TCA cycle) branching from glycolysis, can promote tumorigenesis either alone or in combination [25]. The metabolic intermediates of the TCA cycle play a significant role in the metabolic aberrations in PDAC. Research studies have suggested that the enzymes regulating the metabolic pathways, including glycolysis and the TCA cycle, play a key role in metabolic alteration and thus could be novel therapeutic targets. Mitochondrial malic enzymes (ME2 and ME3) are the oxidative decarboxylases catalyzing malate to pyruvate in the TCA cycle (Figure 1) and are essential for nicotinamide adenine dinucleotide phosphate (NADPH) regeneration and reactive oxygen species (ROS) homeostasis. Dey et al. [26] reported diminished NADPH production and higher levels of ROS in mitochondrial malic enzyme-deficient PDAC cells (PATU8988T). These metabolic changes lead to the activation of AMP-activated protein kinase (AMPK) and suppression of the sterol regulatory element-binding protein 1 (SREBP1)-directed transcription of the branched-chain amino acid transaminase 2 (BCAT2) gene, the gene essential for transferring the amino group from branched-chain amino acids to α-ketoglutarate (α–KG) (Figure 1). This results in the regeneration of glutamate, which in turn takes part to support de novo nucleotide synthesis. These findings support that malic enzyme may serve as a novel therapeutic target in the treatment of PDAC [26]. Recently, the role of paraoxonase 2 (PON2) as a new modulator of glucose transporter 1 (GLUT1)-mediated glucose transport via stomatin has been reported [19]. Knock-down studies in mice have shown that the loss of PON2 is associated with cellular starvation and activation of AMPK that leads to activation of Forkhead Box O3A (FOXO3A) and its transcriptional target, *PUMA*. This results in the induction of anoikis to suppress tumor growth and metastasis of PDAC. The authors suggested PON2 as a new modulator of glucose transport and pharmacologically tractable pathways necessary for tumor growth and metastasis of PDAC, and thus a novel target to attenuate tumor growth and metastasis. Targeting the LDH-A, the enzyme converting the pyruvate to lactate (Figure 1) and which is overexpressed in PDAC, is another potential therapeutic target in the treatment of PDAC. However, the therapeutic efficacy is p53 status-dependent and is favorable with the absence of p53 function as reported by Rajeshkumar et al. [27] using a patient-derived xenograft (PDX) model (6-week-old male nu/nu athymic mice) and an inhibitor of LDH-A (FXII). BAG3, a member of the human Bcl-2–associated athanogene (BAG) cochaperone family, is another potential target in PDAC patients as its aberrant expression significantly contributes to the reprogramming of glucose metabolism via increased expression of hexokinase 2, the first key enzyme in aerobic glycolysis (Figure 1) [28]. Further, the association of mucin 1 (MUC1) and mucin 13 (MUC13) with hypoxia-induced factor-1-alpha (HIF-1α) in PDAC [25] has been documented, and recent findings associate MUC13 with higher glucose metabolism via activation of NF-κB, p65, and phosphorylation of IκB, which result in the upregulation of Glut-1, c-Myc, and Bcl-2 [29], suggesting that MUC13 is an important therapeutic target.

Nuclear factor-kappa beta (NF-kB) and Rel (RelA, RelB, and c-Rel) play crucial roles in the pathogenesis, cancer progression, and apoptosis resistance of PDAC and might be potential therapeutic targets [30,31]. Similarly, the involvement of NF-kB in TRAIL/NF-κB/CX3CL1-mediated oncoimmuno crosstalk resulting in TNFα-related apoptosis-inducing ligand (TRAIL) resistance signifies the role of NF-kB in pancreatic cancer [32]. Further, deregulation of epithelial-mesenchymal transition (EMT) and neural invasion via NF-kB inhibition in pancreatic cancer delineates the possibility of targeting NF-kB in PDAC [33]. PDAC tumorigenesis is associated with NF-kB activation that is epigenetically regulated by long noncoding RNA-*PLACT1*, therefore suggesting that *PLACT1* regulating NF-kB might be a novel therapeutic target [34]. The upregulation of inhibitor of kappa kinase subunit-epsilon (IKKε) in PDAC has been associated with poor clinical outcome. IKKε plays a significant role in the activation of nuclear factor-kappa beta (NF-κB), which also plays a key role in the pathophysiology of PDAC [35]. Aberrant expression of IKKε, even in the absence of gene-amplification, is associated with PDAC via playing a role in glucose metabolism reprogramming. Zubair et al. [35] suggested that the silencing of IKKε in PDAC cells (MiaPaCa and Colo357) is associated with a reduction in malignant cell growth, clonogenic potential, lactate secretion, glucose consumption, metastasis, tumorigenicity, and the expression of genes involved in glucose metabolism. Further, it was also reported that this aberration in metabolism does not impact the basal oxygen consumption rate. These findings suggest that IKKε is a novel therapeutic target in PDAC.

### 3.2. Lipid Metabolism

Elevated fatty acid synthesis is one of the most important metabolic alterations in cancer cell metabolism. Various studies have described the association of altered lipid metabolism and increased expression of enzymes involved in lipid metabolism with PDAC, including using chylomicron-derived fatty acids in addition to lipogenesis [36,37,38]. Thus, lipid metabolism plays a significant role in PDAC pathogenesis. However, the uptake of fatty acids from circulation or reprogrammed lipogenesis for growth and proliferation needs further investigation. Along with the TCA cycle to generate more ATP, the metabolic intermediates produced via the Warburg effect are also funneled into lipogenesis and form short-, medium-, and long-chain fatty acids [13]. The changes occurring within the tumor microenvironment regulate glycolysis. However, changes in the pH or levels of oxygen do not regulate the process of lipogenesis. Oncogenic gene expression increases the expression of the enzymes of lipogenesis such as fatty acid synthase (FASN) (Figure 1), stearoyl-CoA desaturase (SCD1), and acetyl-CoA carboxylase-1 (ACC1) which are involved in lipid synthesis. Lipogenesis and lipid metabolites support tumor growth by facilitating immune system evasion, intercellular signaling, and provide the reducing power in the tumor microenvironment [39]. Since lipogenic gene expression is associated with tumor aggressiveness, targeting the enzymes such as FASN, SCD1, ACC1, and the transcription factor that regulates lipogenic gene expression (sterol-regulatory element-binding protein-1c (SREBP1c)) with lipogenic inhibitors might be of therapeutic importance. Recently, the role of inhibiting the FASN with orlistat in overcoming the gemcitabine resistance has been documented [39]. The potential role of liver-X-receptor (LXR) in cancer metabolism (glycolysis and lipogenesis), progression, and immune evasion has been discussed in the literature. Recently, Flaveny et al. [13] reported the possibility of targeting the nuclear liver-X-receptors—LXRα, and LXRβ (NR1H3 and NR1H2, respectively)—which are the key regulators of lipid, cholesterol, and carbohydrate metabolism and homeostasis. The authors documented the role of LXR inverse agonist, SR9243, which inhibits LXR activation by enhancing LXR-corepressor recruitment. This results in reduced cancer cell viability, enhanced apoptotic cell death, cancer cell sensitization to chemotherapy, disruption of the Warburg effect, and suppression of lipogenesis gene expression and growth of the tumor. SR9243 was found to be nontoxic to normal cells. However, its activity in promoting immune-mediated tumor clearance needs to be studied and further research is needed for definitive therapies with minimal adverse side effects [13].

Desmoplasia is associated with the tumorigenicity and malignant progression of PDAC as well as reinforces the proliferating tumor to metabolically adapt the tumor microenvironment. Guillaumond et al. [38], using male Pdx1-Cre, Ink4a/Arf^fl/fl^, and LSL-Kras^G12D^ mice which develop spontaneous PDAC, reported that aberration in lipid metabolism and lipid-related metabolic pathways, mainly lipoprotein metabolic processes and cholesterol uptake, is considerably activated in PDAC. This increases the cholesterol amount and overexpression of the low-density lipoprotein receptor (LDLR) in tumor cells. The silencing of LDLR with short hairpin ribonucleic acid (shRNA) was associated with an attenuated cholesterol uptake, decreased tumor cell proliferation and clonogenic potential, and limited activation of ERK1/2. Furthermore, there is an association of higher risk of disease recurrence with increased LDLR expression and increased sensitization of tumor cells to chemotherapy with blocking cholesterol uptake, thus elucidating the key role of lipid metabolism and LDLR in the metabolic aberration. The authors suggest LDLR as a novel therapeutic target in the treatment of PDAC.

### 3.3. Amino-Acid and Nucleotide Metabolism

The key role of various amino acids including glutamine and their metabolomics and metabolic alterations are under research for elucidating better therapeutic targets to treat PDAC. Glutamine is an essential amino acid used in the biosynthesis of proteins and plays a key role in nucleotide synthesis in PDAC. Thus, the metabolic alterations in glutamine metabolism may enhance tumor growth and proliferation [18]. Glutamate ammonia ligase mediated glutamine biosynthesis fuels the TCA cycle by being deaminated via glutaminolysis to α-ketoglutarate (aKG) and couples the TCA cycle with nitrogen anabolism, playing a critical role in PDAC (Figure 1) [40]. Further, the growth of pancreatic cancer through a Kras-regulated metabolic pathway in association with glutamine supports the role of glutamine in PDAC [16]. Mouse models, cell lines derived from pancreatic cancer of mice or humans, and xenograft models are being used to study the amino acid metabolomics. The results of these studies might have translational aspects but cannot be directly applied to humans given that different tissues from different animals may behave differently. Mayers et al. [41] reported unique metabolism of the branched-chain amino acids in different tumor tissues of the same pathological origin from humans and mice. These findings suggest that researchers should be more vigilant in choosing the model of study and suggest the need for more clinical trials for PDAC treatment.

The switching of glucose to glutamine metabolism in MUC1 overexpressing cells during glucose deprivation has been reported in PDAC cells. This switching to glutamine metabolism was associated with decreased cell proliferation, G1 phase arrest, decreased DNA synthesis, and disruption in pyrimidine nucleotide biosynthesis, leading to a significant accumulation of glutamine-derived aspartate in MUC1 overexpressing cells (S2−013) [42]. Abrego et al. [18] reported that, in low-pH conditions, PDAC cells deviate from glycolytic metabolism and rely more on oxidative metabolism facilitated by increased expression of the aspartate aminotransferase (GOT1) enzyme (Figure 1) and enhance the non-canonical glutamine metabolic pathway. The results from this study suggest the key role of *GOT1* in PDAC cell (S2−013 and Capan-1) survival in chronic acidosis and anaplerotic glutamine metabolism as a key therapeutic target for PDAC treatment. Thus, targeting the glutamine metabolism might serve as novel therapeutics for the treatment of PDAC.

Clinical trials with CB-839 have been conducted with the pharmacological inhibition of glutaminase (GLS) (Figure 1); however, there is a lack of optimal safety and efficacy. CB-839 is a potent GLS2 inhibitor but not for liver-expressed GLS2. Since GLS inhibition may decrease energy production, the glutamine pathway seems to be a potential therapeutic target. However, a sustained proliferation of pancreatic cancer cells in vitro and in vivo after GLS inhibition despite marked early effects on in vitro proliferation, suggests the presence of alternative routes or compensatory mechanisms for acquiring the energy required for growth and proliferation [43]. Biancur et al. [43] reported the presence of compensatory mechanisms or adaptive responses after targeting GLS and suggested that these compensatory mechanisms should be targeted along with GLS inhibition for better results. Elgogary et al. [44] reported the enhanced efficacy and improved pharmacokinetics of encapsulated bis-2-(5-phenylacetamido-1,2,4-thiadiazol-2-yl) ethyl sulfide (BPTES) nanoparticles, a selective glutaminase inhibitor, using a PDx mouse model (four-week old female Foxn1^nu^ athymic nude mice). BPTES nanoparticle monotherapy showed a modest antitumor effect by acting only on the proliferating tumorous cells. Further, there is greater tumor reduction with the combined therapy of BPTES nanoparticles with metformin to target glutamine and glucose metabolism together, suggesting that targeting multiple metabolic pathways is more beneficial. However, Chaiteerakij et al. [45] reported that there is no survival benefit with metformin in PDAC patients. Thus, further research is needed to ascertain the clinical efficacy of this combination in the survival of the patients. The role of glutamine use by tumors and the effectiveness of targeting glutamine metabolism is a current debate in the field. The heterogeneity of glutamine metabolism in-vivo and ex-vivo and the role of glutamine in epigenetic regulation and post-translation modification via regulating acetylation and demethylation depicts its importance [46]. Moreover, despite understanding the dependency of cancer cells on glutamine for growth and survival under physiological conditions, the contribution of glutamine metabolism to tumor progression remains under investigation, partially because of the low glutamine levels found in the tumor environment [47]. It is also important to understand how pancreatic tumor cells respond and adapt to glutamine starvation to optimize glutamine targeted therapy.

Increased tumor growth and mass depends on the increased protein synthesis, which in turn requires increased nucleotide synthesis. For continued growth, tumor cells depend on metabolic alteration for increased energy needs and nucleotide synthesis. To fuel the increased anabolic process, PDAC cells depend on the amino acid glutamine. Son et al. [16] reported that the metabolic reprogramming of glutamine metabolism is essential for tumor growth. The group reported that PDAC cells (8988T) use the non-canonical glutamine metabolic pathway where glutamine-derived aspartate is converted into oxaloacetate by aspartate transaminase (GOT1), followed by malate and pyruvate to maintain the redox state. This is contrary to normal cells in which glutamate dehydrogenase (GLUD1) is needed to convert glutamine-derived glutamate into α-ketoglutarate to fuel the TCA cycle (Figure 1). The authors also reported that alteration of any enzyme in this pathway results in increased ROS. Thus, the enzymes involved in this pathway might serve as novel therapeutic targets. Increased pyrimidine biosynthesis enhanced the intrinsic levels of deoxycytidine triphosphate (dCTP) by facilitating MUC1-mediated HIF-1α stabilization (Figure 1) and induced phosphoribosyl pyrophosphate in gemcitabine-resistant PDAC cells (Capan-1, T3M4, and MIA PaCa-2), thus delineating the role of nucleotide metabolism and metabolic alterations in treatment resistance [48]. Moreover, *KRAS*-driven metabolic adapting cancer cells grow and proliferate depending on the altered metabolism by up-regulating key enzymes of amino acid, fatty acid, or nucleotide biosynthesis, and stimulation of scavenging pathways (macropinocytosis and autophagy) [49]. Further, increased nucleotide synthesis is mediated by *MYC* upregulation, the transcription of ribose-5-phosphate isomerase (*RPIA)*, the non-oxidative pentose phosphate pathway (PPP) gene, via KRAS dependent mitogen-activated protein kinase (MAPK)-dependent signaling pathway activation, and the inhibition of the growth of KRAS-resistant cells by antagonizing the PPP or pyrimidine biosynthesis [50], supports the notion of targeting nucleotide (purine and pyrimidine) metabolism in PDAC. Similarly, the role of proline, glycine, alanine, aspartate, asparagine, purine, pyrimidine, and branched-chain amino acid (BCAA) in PDAC metabolism and as therapeutic targets has been documented [51,52,53].

Pancreatic cancer shows increased expression of proline oxidase (PRODH1). The enzyme mediates proline-derived glutamate (Figure 1) and promotes survival and proliferation under glucose- or glutamine-limited conditions [54]. The study suggests that collagen-derived proline promotes survival and proliferation through TCA cycle metabolism and cellular respiration. However, investigating the collagen-derived free proline to protein synthesis as a mechanism providing energy will be of interest. The clonogenic and tumorigenic potential of various cell lines (including pancreatic cancer) correlate with proline consumption, expression of proline biosynthesis enzymes, and inhibition of proline biosynthesis enzymes or proline starvation; this association with proline is linked with impaired clonogenic or tumorigenic potential [55]. The role of proline in tumor growth and metastasis has been discussed in the literature (reviewed in [56,57]). The energy needs of the tumor cells can also be fueled by the alanine secreted by stroma-associated pancreatic stellate cells (PSCs). PSC-derived alanine secretion (Figure 1) is dependent on PSC autophagy, the secreted alanine to fuel the tricarboxylic cycle (alanine to pyruvate by alanine transaminase), and non-essential amino acids and lipid biosynthesis. Thus, alanine as an alternative fuel source promotes growth in an austere tumor microenvironment [58].

Recently, it has been demonstrated that SLC1A4 transporters are used by PSCs to rapidly exchange and maintain environmental alanine concentrations and upregulation of SLC38A2 by PDAC cells (PANC1 and MiaPaCa2); they also facilitate the alanine supply to meet the energy demands [59]. Another potential target in pancreatic tumors might be asparagine. Asparagine synthetase (ASNS) converts aspartate and glutamine to asparagine and glutamate (Figure 1), and low or subdetectable levels of ASNA were found in 70% of 98 PDAC patient samples [60]. This suggests that some patients may be susceptible to ASNase therapy. Moreover, it has been documented that overexpression of ASNS under glucose-deprived conditions protects pancreatic cancer cells from apoptosis induced by glucose deprivation and cisplatin [61]. Asparagine restriction in pancreatic cancer cells activates receptor tyrosine kinase-MAPK signaling and enhances translation of activating transcription factor 4 (ATF4) mRNA. MAPK inhibition attenuates translational induction of ATF4 and the expression of its target ASNS, as well as sensitizes the pancreatic tumor to asparagine restriction; this is reflected by tumor growth inhibition. Further, low ASNS expression is among the top predictors of responsiveness to MAPK inhibition in melanoma [62]. These results suggest that a combinational approach of MAPK inhibition with asparagine restriction should be investigated as a therapeutic approach in selected PDAC cases with low-ASNS. In mice, early stages of mutant *Kras* driven pancreatic cancer are associated with increased levels of BCAAs (originated from breakdown of tissue protein). In humans, increased levels of BCAAs are associated with an increased risk of developing pancreatic cancer [63] and early events of pancreatic cancer [63,64]. However, PDAC tumors have decreased BCAA uptake at later stages [41]. Lee et al. reported an increased uptake of BCAAs by PDAC cells (8988 T and MIAPACA2) and the important role of BCAA metabolism in human PDAC growth by regulating lipogenesis, as well as demonstrated that BCAAs are the source of carbon (Figure 1) for fatty acid synthesis [52]. Thus, BCAA concentration may serve as an early predictor of PDAC and the metabolic pathways may be novel therapeutic targets.

### 3.4. Mitochondrial Metabolites and Reactive Oxygen Species

In the tumor cells, activation of oncogenes, such as MYC and KRAS, and deregulation of signaling pathways, including the PI3K pathway, results in increased glycolysis. This leads to the generation of glycolytic and TCA cycle intermediates [65]. The funneling of these intermediates into the pentose phosphate pathway for NADPH, generation of more ATP, and nucleotide production aids in cell proliferation and growth of the tumor [66]. TCA cycle intermediates can also be generated by the stepwise oxidation of glutamine to generate α-ketoglutarate metabolizing further to oxaloacetate, acetyl coenzyme A, and citrate. These metabolic aberrations lead to tumor addiction to glucose and/or glutamine. Increased oxidative metabolism results in increased generation of ROS by mitochondria, which in turn activates the signaling pathways proximal to the mitochondria and promotes cancer cell proliferation and tumorigenesis [15]. However, the accumulation of ROS is detrimental to the tumor cells and may lead to necrosis. Tumor cells generate an increased amount of NADPH to counter the increased ROS. Thus, both glucose-dependent metabolic pathways and mitochondrial metabolism are essential for tumor growth. Comparable results were reported by Weinberg et al. [67] using an oncogenic Kras-driven mouse model (LSL-Kras G12D mice on a mixed B6/129 background) of lung cancer. The authors reported that the major function of glucose metabolism is to support the pentose phosphate pathway, glycolytic ATP is to support growth under hypoxic conditions, and that alpha-ketoglutarate, the tricarboxylic acid cycle intermediate via glutamine metabolism and ROS generated from mitochondrial metabolism, is essential for Kras-induced anchorage-independent growth. Further, decreased tumorigenesis after disruption of the mitochondrial function supports the idea of targeting the mitochondrial metabolism to decrease cell proliferation and tumorigenesis in a Kras-dependent tumor. Furthermore, disrupted mitochondrial function associated with PDAC and reduced mitochondrial oxidative phosphorylation (OXPHOS) by enhancing mitochondrial fusion, through dynamin-related protein-1 (Drp1) inhibition or mitofusin-2 (Mfn2) overexpression, is associated with tumor growth suppression and improved survival. Thus, supporting the targeting of mitochondria as a therapeutic target [68].

Increased number of mitochondria in tumor tissue facilitates tumor growth. Franco et al. [14] reported that CDK4/6 inhibition is associated with an increased number of mitochondria, ROS production, and changes in cellular metabolism. However, using a combination therapy of CDK4/6, MEK/mTOR inhibitors, and targeting the mitochondrial enzymes, including homoxygenase-1 and catalase, results in decreased tumor growth and a more durable therapeutic response. Thus, metabolic reprogramming by inhibiting CDK4/6 and targeting the mitochondrial enzymes might be a promising strategy for PDAC treatment. Activation of MAPK, ERK1/2, and NF-κB for cellular proliferation; c-SRC, NF-κB, PIK3/AKT, and JAK/STAT for apoptosis evasion; MMPs for tissue invasion and metastasis; release of vascular endothelial growth factor (VEGF) and angiopoietin for angiogenesis; and c-MET overexpression and Ras homolog gene/Ras-related C3 (RHO-RAC) interaction mediated by ROS for progression and aggressiveness of PDAC has been discussed [69]. Further, initiation of premalignant PDAC by ROS is regulated by TIGAR protein, while restricting metastasis suggests an ROS-mediated PDAC phenotypic switch by increasing migration, invasion, and metastatic capacity. In brief, PanINs and tumor proliferation is associated with higher expression of TIGAR and low levels of ROS. Contrastingly, metastasis and invasion are correlated to higher levels of ROS, decreased expression of TIGAR, and activated phospo-extracellular signal-regulated kinase (pERK). The activated pERK suggests that phenotypic switch is MAPK and TIGAR level dependent [70]. Further, the reversal of activated MAPK with antioxidant suggests the ROS-TIGAR-MAPK axis is a potential therapeutic target for controlling tumor progression. Targeting mitochondrial proteins and ROS levels in PDAC is also supported by the findings that increased fission and mitochondrial ROS generation is associated with silencing of the family with sequence similarity 49 member B (FAM49B-a mitochondrial protein regulating mitochondrial fission and cancer progression) in PDAC cells (CFPAC1, MIAPaCa2, BxPC3, PANC1) and enhanced proliferation and invasion [71]. These findings support that overexpressing FAM49B might reduce proliferation and invasion by attenuating ROS production.

Moreover, the relation between mitochondrial aspartate transaminase (GOT2) with the cellular redox state (ROS production), senescence, and PDAC growth suppression [72] supports the notion of targeting ROS-associated pathways in PDAC. These findings suggest the dual role of ROS in pancreatic cancer. At moderate levels, ROS facilitates proliferation, growth, apoptosis, and invasion, while, at low or high levels, ROS retards tumor cell growth and leads to cell death. Thus, ROS can be targeted in pancreatic cancer [73,74]. Recently, Shao et al. [75] reported a positive feedback loop in Caveolin-1-ROS signaling in PSCs leading to increased expression of glycolytic enzymes (hexokinase 2 (HK-2), 6-phosphofructokinase (PFKP) and pyruvate kinase isozyme type M2 (PKM2)), upregulated transporter (Glut1) expression, and downregulated expression of oxidative phosphorylation (OXPHOS) enzymes (translocase of outer mitochondrial membrane 20 (TOMM20) and NAD(P)H dehydrogenase (quinone) 1 (NQO1)) resulting in a shift in energy metabolism to glycolysis, which promotes the growth of the tumor. Thus, interrupting this pathway or metabolic coupling between the stroma and tumor cells may be an effective therapy. Further, the adaptation of PSCs to oxidative stress induced by hypoxia is associated with an increase in mitochondrial ROS and increased oxidation of proteins and lipids suggests that ROS plays a role in tumor growth and viability [76]. Hypoxia is associated with increased expression and activity of superoxide dismutase, decreased oxidized/reduced glutathione ratio, increased phosphorylation of nuclear factor erythroid 2-related factor, and increased expression of antioxidant enzymes catalytic subunit of glutamate-cysteine ligase, catalase, NAD(P)H-quinone oxidoreductase 1, and heme oxygenase-1 [76]. Targeting ROS in PDAC is also supported by the fact that ROS induces monocyte-to-myofibroblast transdifferentiation. These transdifferentiated fibroblasts induce reactive stroma and promote tumor progression [77]. Measuring the ROS levels and mitochondrial membrane potential may serve as an indicator of epithelial-mesenchymal transition [78], a causative factor inducing resistance [79]. Recently Olou et al. [80] reported a novel mucin1-cytidine deaminase (MUC1-CDA) axis of the adaptive UPR facilitating the survival advantage upon ER stress induction as well as documented the roles of MUC1 in protecting against ROS insults and the antioxidant property of deoxyuridine in modulating ROS levels. These results suggest that decreasing ROS levels has a therapeutic effect. However, Chattopadhyay et al. [81] reported inhibited growth of MIA PaCa-2 and BxPC-3 pancreatic cancer cells with NOSH-aspirin with increased ROS levels. Further, in a xenograft mouse model, NOSH-aspirin decreased the tumor growth associated with increased expression of ROS, iNOS, and mutated p53 while NF-κB and FoxM1 expression were inhibited.

## 4. Autophagy and Macropinocytosis

Autophagy plays a crucial role in sustaining the continued growth of pancreatic cancer by recycling the cell organelles and protein aggregates through autophagosomes. The autophagosomes are further degraded in lysosomes to provide the metabolic intermediates to support increased ATP production [4]. Autophagy plays a crucial role in tumor stem cell maintenance, tumor cell migration, and invasion, therapy resistance, regulating the interaction of cancer cells with other cell types in the tumor microenvironment (reviewed in [82]). Autophagy plays a context-dependent role in pancreatic cancer and has both a pro-tumorigenic and antitumorigenic role. Autophagy suppresses primary tumor growth but also plays an important role in the initiation of tumorigenesis, tumor maintenance, tumor cell invasion and motility, progression, cancer stem cell differentiation, escape from immune surveillance, and metastasis. The anti-tumorigenic role of autophagy is demonstrated by tumor inhibition with Beclin-1 and increased rates of tumorigenesis in mice heterozygous for the autophagy gene AMBRA-1. The conflicting role of autophagy involving Atg3, Atg5, Atg7 genes, p62 proteins, and MAPK and NF-kB pathways and positive (induction of Vacuole membrane protein 1, small nucleolar RNA host gene 14, and osteopontin) and negative (UBL4A Lysosome-associated membrane protein-1, optineurin, and precursor of nerve growth factor) regulators of autophagy in PDAC development and its impact on metabolism has been reviewed elsewhere [83,84].

TGFB1 induced autophagic flux mediated increased proliferation and inhibition of migration in SMAD4-positive PDAC cells by decreased nuclear translocation of SMAD4; and inhibition of proliferation and increased migration in SMAD4-negative cells through the regulation of MAPK/ERK activation further supports the context-dependent role of autophagy and the probable role of autophagy in PDAC heterogeneity [85]. Lumican is a class II small leucine-rich proteoglycan with a key role in ECM organization. The presence of lumican in ECM surrounding PDAC inhibits cancer cell replication and favors an improved outcome. However, hypoxia-induced autophagy-mediated degradation and reduction in protein synthesis within PSCs decrease stromal lumican. These findings suggest the plausible mechanism of hypoxia-induced autophagy in tumor growth and autophagy inhibition as a potential therapeutic mechanism [86]. Autophagy maintains tumor growth by both intrinsic and extrinsic cellular mechanisms and the significant regression of tumor growth after inhibition of autophagy in an acute and reversible autophagy inhibition mouse model (ATG4B^CA^) suggests that targeting autophagy has therapeutic utility in PDAC [87]. Further, a cell-autonomous and non-cell-autonomous pro-tumorigenic role for autophagy in PDAC is supported by the involvement of stromal and immunologic mediators contributing to the antitumor effects of autophagy inhibition. This provides strong evidence to the effectiveness of acute and potent inhibition of autophagy in treating pancreatic cancer in mouse models. ERK inhibition sensitizes PDAC to autophagy inhibition by hydroxychloroquine and leads to a synergistic decrease in tumor cell proliferation, tumor growth, and enhanced apoptosis. Recently, increased autophagic flux with suppression of KRAS as with pharmacological inhibition of its effector ERK MAPK and decreased glycolytic and mitochondrial functions with KRAS suppression or ERK inhibition was reported. Additionally, the synergistic effect of autophagy inhibition with chloroquine inhibition of specific autophagy regulators genetically or pharmacologically in enhancing the ability of ERK inhibitors to mediate antitumor activity in KRAS-driven PDAC [88] supports the combinational approach of blocking ERK MAPK and autophagy as an effective treatment for PDAC.

Autophagy can be suppressed by melanoma-associated antigen family A (MAGEA6) in PDAC cells (AsPC-1, BxPC-3, Capan-1, Capan-2, Panc-1, and MIA PaCa-2) and this suppression is released upon MAGEA6 degradation induced by nutrient deficiency or by the acquisition of cancer-associated mutations. Tsang et al. [89] reported tumor initiation with inhibition of autophagy and tumor progression with the reinstitution of autophagy in a nude female mouse model. These findings provide mechanistic insight into the divergent roles of MAGEA6 regulating autophagy and MAGEA6 as an immunotherapeutic target for PDAC. Further, immune evasion in PDAC is facilitated by MHC-I degradation and this process is mediated by autophagy, thus autophagy inhibition should be considered with immunotherapy to get a synergistic effect [90]. Gemcitabine resistance in BxPC-3 and Capan-1 cells (CD44+ cells) with higher autophagy activity was related to higher expression of *ATG12, UVRAG, MAP1LC3B*, and *MAP1LC3A* genes, and suppression of these autophagy regulatory genes might be a potential therapeutic strategy [91].

Macroautophagy plays a critical role in maintaining energy homeostasis and glutamine metabolism, which in turn is required for PDAC tumor survival [92]. Glutamine deprivation is associated with activation of macropinocytosis-associated autophagy via activation of transcription factor EB. Glutamine deprivation is also associated with activated apoptotic cell death upon autophagy inhibition. However, it is not associated with the induction of cell death. Autophagy inhibition is associated with increase glutamine uptake to compensate for decreased intracellular glutamine [92]. These findings suggest autophagy as a potential therapeutic target. In the presence of altered metabolism and high energy demands, PDAC relies on lysosome-dependent recycling pathways to engender metabolic substrates. Autophagy and micropinocytosis, mediated by activated and functional lysosomes in PDAC, are used to acquire nutrients for tumor cell growth and proliferation and are associated with the suppression of tumor growth. The inhibition of lysosome function is also associated with tumor growth suppression. Further, the co-dependency of functional lysosomes and replication stress response by aspartate depletion has been documented [53]. Recently, Zhao et al. [93] proposed MEK inhibition and targeting lysosomal function together to improve sensitivity to MEK inhibition in PDAC. This was based on the findings of a time-dependent increase in lysosomal content of MEK inhibitor trametinib or refametinib associated with nuclear translocation of the Transcription Factor EB; the mechanism behind the resistance to MEK inhibition. A similar treatment strategy to combine MEK inhibition and targeting autophagy was proposed by Kinsey et al. [94] and the group reported that MEK1/2 inhibition leads to LKB1→AMPK→ULK1 signaling axis activation and regulate autophagy. Anti-proliferative effects in PDAC cells (Mia-PaCa2 and BxPC3) and regression of xenografted patient-derived PDA tumors in mice noticed by combined inhibition of MEK1/2 and autophagy suggests synergism of autophagy and MEK inhibition.

Pancreatic tumors also acquire nutrients from ECM in the form of collagen and collagen-derived proline to fuel growth and proliferation. Olivares et al. [54] reported that glucose or glutamine deprivation promotes uptake and breakdown of ECM collagen proteins I and IV by PDAC cells (PK4A). Following glucose starvation under normoxia, macropinocytosis is used for collagen uptake. While collagen uptake in PDAC cells following glutamine starvation or hypoxia is macropinocytosis-independent. The growing nutrient needs of the tumor are fueled by a highly conserved endocytic process called macropinocytosis, which facilitates the internalization of extracellular fluid and its contents such as amino acids and proteins into cells via macropinosomes. The internalized proteins are only utilized by the PDAC tumorous cells and not by the normal cells [95,96]. Activating K-ras mutation occurs in > 90% PDAC [4] and oncogenic Ras proteins enhance the acquisition of extracellular nutrients to fuel the unmet energy needs of the tumor through the process of micropinocytosis [95]. Ras-transformed cells that are highly dependent on glutamine for growth have been shown to utilize micropinocytosis to fuel the increased nutrient needs by internalizing the extracellular proteins into the cells which undergo proteolytic degradation to yield amino acids including glutamine. These amino acids can enter into the central carbon metabolism to yield ATP, fulfilling the increased energy needs [95]. Further, decreased amino acids within the tumor, and compromised growth of Ras-transformed pancreatic tumor xenografts after pharmacological inhibition of macropinocytosis with 5-N-ethyl-N-isopropyl amiloride (EIPA), suggests that macropinocytosis is a novel therapeutic target [95,96]. These results suggest that investigating autophagy, micropinocytosis, macropinocytosis, lysosomal activity, and transport pathways in greater depth might uncover better therapeutic targets.

## 5. Aminoacid and Glucose Transporters

Increased protein synthesis requires increased amino acid synthesis and/or transport within the cell. Amino acid transporters serve to flux the amino acids and hence might be targeted to decrease the amino acid availability for protein synthesis. The upregulation of amino acid transporter SLC6A14 has been reported in pancreatic cancer by Coothankandaswamy et al. [97]. The findings of amino acid starvation in pancreatic cancer cells are associated with reduced growth and proliferation after pharmacological inhibition of SLC6A14. This suggests that amino acid transporters could be potential therapeutic targets for suppressing the growth of the tumor. SLC1A4 and SLC38A2 (alanine transporter) [59] and SLC1A5 (ASCT2-glutamine transporter) [98] enhance PDAC tumor growth by importing essential amino acids and therefore represent additional potential therapeutic targets. The association between increased large neutral amino acid transporter 1 (LAT1-SLC7A5), expression with chemo-resistance [99] and poor prognosis [100] in PDAC suggests LAT1 be a potential therapeutic target. LAT-1 provides a conduit to acquire amino acids, like leucine and methionine, into the tumor tissue. Inhibition of leucine and methionine import and the decreased intracellular pools of methionine, leucine, and polyamine with LAT1 inhibitors indicates that LAT1 is a druggable target. The enhanced efficiency of piperazine-based materials to prepare LAT1 inhibitors associated with decreased essential amino acid pools supports the hypothesis of targeting LAT1 in PADC [101]. Higher expression of glucose transporter GLUT-1, contributing to the invasiveness and metastasis, is associated with poor prognosis and decreased therapeutic response to neoadjuvant chemotherapy [102]. The findings that PON2 enables growth and metastasis of pancreatic cancer by GLUT1-mediated increased glucose transport [19] suggests that GLUT1 is a therapeutic target. Similarly, the association of PD-1 + T cells with increased expression of GLUT-1 in PDAC makes PD-1 + T cells potential targets for immunotherapy [103]. Upregulated expression of lactate transporter MCT4 in PDAC has been associated with glycolytic metabolism as well as poor survival and outcome. Further, it was also reported that MCT4 has a differential expression in PDAC cell lines. The knock down of MCT4 attenuates the lactate efflux as well as its synthesis from glucose but with persistent glucose uptake. However, this effect was differential and depends on the expression levels of MCT4 in different PDAC cell lines and specimens. MCT4, a key marker and determinant of the metabolic state in aggressive cancer, with its differential expression and dramatic effect on the metabolism with MCT4 depletion (enhanced oxidative phosphorylation of glucose and glutamine, inhibition of pyruvate carboxylation, increased ROS production), suggests that MCT4 is an important mediator for regulating the metabolism and a novel therapeutic target [104].

## 6. Metabolic Alterations, Stemness, and Chemo-Resistance

Stemness is defined as stem-cell-like phenotypes of cancer cells. Cancer stem cells (CSCs) have the properties of self-renewal, differentiation, and proliferative potential. As CSCs differentiate, they lose the stemness [105,106]. Stemness facilitates the growth and metastasis of the tumor, and the survival of chemoresistant CSCs leads to chemoresistance. Zhang et al. [107] reported that gemcitabine enhances the ratios of CD24+ and CD133+, the CSC markers, as well as the expression of Bmi1, Nanog, and Sox2, the genes associated with stemness. This leads to increased cell migration, chemoresistance, and tumorigenesis. The authors also documented that the gemcitabine-induced resistance is mediated by the NADPH oxidase/ROS/NF-κB/STAT3 pathway and thus this pathway might serve as a potential therapeutic target. Gemcitabine resistance may also be due to combined effects of C-X-C motif chemokine receptor 4 (CXCR4) and hedgehog pathways via bidirectional tumor-stromal crosstalk. Khan et al. [108] reported that this resistance can be abrogated by inhibition of the CXCR4 and hedgehog pathways using pancreatic cancer cells (MiaPaCa and Colo357). The study reported reduced tumor growth with gemcitabine alone or in combination with a CXCR4 antagonist (AMD3100) or hedgehog inhibitor (GDC-0449) in orthotopic pancreatic tumor-bearing mice. This suggests that triple therapy is more effective for nearly complete suppression of tumor growth.

Poor or inadequate response to the therapeutic agents by the tumor tissue is a well-known phenomenon in PDAC. Gemcitabine resistance is a challenge for the clinicians in treating locally advanced and metastatic pancreatic cancer. FOLFIRINOX has also been used in such patients. However, to use FOLFIRINOX (FFX) or gemcitabine plus nab-Paclitaxel (GN) in such patients remains debatable. Studies support the use of FFX followed by GN if FFX fails [109], preferably GN [110], and FFX followed by GN or GN followed by FFX with equal outcome [111]. This might be due to the scarcity of data comparing the sequential use of these drugs. Metabolomic alterations of the tumor cells might play a role in treatment resistance or decreased response to gemcitabine and, therefore, ongoing research being conducted to overcome this problem. One strategy might be to increase the sensitivity of the PDAC cells to gemcitabine. Recently, Shukla et al. [48] reported the role of MUC1 and HIF-1α in imparting the gemcitabine resistance by increased glycolytic flux leading to glucose addiction in cancer cells. The authors reported that this metabolic alteration leads to an increase in pyrimidine biosynthesis (Figure 1), enhancing the intrinsic levels of deoxycytidine triphosphate (dCTP), which in turn diminishes the effectiveness of gemcitabine via molecular competition and results in a poor response. The authors proposed MUC1, HIF-1 α, and reducing the de novo pyrimidine synthesis as novel therapeutic targets in PDAC.

Metabolic alterations in lipid metabolism have been correlated with poor therapeutic responses, chemoresistance to gemcitabine, and stemness of the tumor. Tadros et al. [39], using athymic female nude mice (NCr-nu/nu) with orthotopic implantation of PANC-1 cells, reported that significantly increased expression of FASN correlates with an increase in disease progression in the spontaneous pancreatic cancer mouse model, poor survival in patients, and poor gemcitabine responsiveness in cell lines. The authors reported that gemcitabine resistance can be overcome by targeting lipogenesis by inhibiting FASN (Figure 1) with orlistat, in part due to endoplasmic reticulum (ER) induction. Similar effects of decreased stemness and synergistic activity with gemcitabine were noticed with thapsigargin by direct ER stress induction. The overexpression of oncogenes MUC1 and MUC13 have been well established in many solid tumors including pancreatic cancer. Further, the role of MUC1 in facilitating the growth and proliferation, promoting metabolic alterations, reducing the overall survival, and imparting the resistance to radiation and chemotherapies in PDAC has also been documented. Gunda et al. [5] reported the role of MUC1-mediated metabolism in imparting the resistance to radiation therapy in PDAC and found that MUC1 decreased the radiation-induced cytotoxicity and DNA damage in pancreatic cancer cells (S2−013 and Capan2) by enhancing glycolysis, pentose phosphate pathway, and nucleotide biosynthesis (Figure 1). The results suggested that MUC1-mediated metabolic aberrations in nucleotide synthesis facilitates the radiation resistance and thus MUC1 might be a novel therapeutic target. Chemoresistance to gemcitabine, the only FDA-approved drug for PDAC, results in poor clinical outcomes in pancreatic cancer. Chemoresistance to the drugs is due to the presence of cancer stem cells (CSCs) and the cells responsible for epithelial-mesenchymal transition (EMT) [79]. CSCs endowed stemness to cancer cells, the capacity of cancer cells for self-renewal. Stemness of the tumor imparts the chemoresistance, tumor recurrence, and metastasis to PDAC. Zhao et al. [79] reported that the metabolic aberration mainly enhanced glycolysis, promoted the expression of doublecortin-like kinase 1, the marker for the stemness, and maintained the CSC and EMT phenotypes in the gemcitabine-resistant cancer cells via maintenance of low-levels of reactive oxygen species. This data suggest that the stemness of the cancer cells is due to the aberrant metabolism and the glycolysis-ROS-DCLK1 pathway, which may be potential targets in PDAC treatment.

The stroma of the tumor is a major obstacle in pancreatic tumor treatment as it attenuates drug delivery to the tumor and leads to drug inefficiency and resistance. Tumor stroma makes more than 2/3 of the tissue; therefore, it is important to focus on the histomorphology and pathology of the stroma to investigate novel therapeutic strategies. The stroma of the tumor provides physical support to the mutated epithelial cells and is a modulator and driver of tumorigenicity. The tumorigenicity, malignant initiation, and progression in pancreatic cancer are due to the presence of a heterogeneous group of connective tissue cells (fibroblast-like cells) termed carcinoma-associated fibroblasts (CAFs) [112]. Stromal cells expressing CD36 have an antitumorigenic effect. However, stromal cells lacking CD36 expression promotes tumorigenesis by mediating a metabolic shift from mitochondrial oxidative phosphorylation to glycolysis, resulting in increased expression in monocarboxylate transporter 4 (MCT4). This suggests that CAFs play a key role in the metabolic shift of the tumor. Thus, targeting CAFs to revert this switch to attenuate the MCT4 expression might be a novel metabolomic target in the treatment of PDAC by decreasing the drug resistance due to epithelial-stromal metabolic coupling, leading to the induction of aerobic glycolysis and proliferation in fibroblasts via oxidative stress [112].

Resistance to the therapy in mesenchymal phenotype pancreatic carcinoma is mediated by tumor-associated macrophages (TAMs) via the process of EMT and thus a potential therapeutic target [113,114]. TAMs are an integral part of the tumor microenvironment (TME), and TME is central to all stages of cancer development including tumorigenesis, progression, and metastasis. TME is comprised of cancer-associated fibroblasts (CAF) and non-cancerous components including adipocytes, endothelial cells, fibroblasts, and immune cells including T cells, B cells, dendritic cells, and macrophages [115]. TAMs induced by the changing tumor microenvironment assimilate the levels of hypoxia and lactate. This leads to the activation of MAPK signaling and induction of macrophages expressing arginase 1 (ARG1) and mannose receptor C type 1 (MRC1). These phenotypic changes in the tumor milieu induce angiogenesis in order to enhance the blood perfusion and nutrient supply in the hypoxic and nutrient-deprived tumor. Carmona-Fontaine et al. [116], using a mouse model of breast cancer, reported that metabolic aberrations in the microenvironment of the tumor result in the heterogeneity of the macrophage population expressing ARG1 and MRC1. The group suggested that the gradient in the metabolites of the tumor act as tumor morphogens. Increased expression of Arg1 and IL-4, with alternatively activated macrophage infiltration in a KRAS^G12D^ p53^R172H^ Pdx-Cre^+/+^ (KPC)-mouse model of cachexia, was found to have a negative correlation between CD163-positive macrophage infiltration and cachexia phenotype in mice, thus suggesting the role of TAMs (via increasing expression of ARG1) and ARG1 in the pathogenesis of PDAC and PDAC associated cachexia [117,118]. Further, TAM-associated angiogenesis augments the extravasation of tumor cells out of blood vessels, induces the epithelial-to-mesenchymal transition, and elevates the glycolytic gene transcript levels [119]. Thus, targeting the metabolic aberrations in tumor morphogens (CAFs) to reprogram the metabolism in macrophages and modulating the expressions of ARG1, MRC1, and MAPK signaling might lead to the discovery of novel therapeutic strategies [116]. Since TAMs promote PDAC progression and chemotherapy resistance, elucidating TAM heterogeneity and functions in depth and focusing on their metabolism could provide insight for better therapeutics [120,121,122]. Though TAMs have protumor properties, rewiring the metabolism of TAMs and circulatory macrophages might be of therapeutic importance. The acquisition of the anti-tumor activity by macrophages through activated CpG, a TLR9 agonist, via changes in the central carbon metabolism of macrophages supports this notion [123]. Additionally, blocking the pro-metastatic TAM phenotype by treating PDAC TAMs with 2-deoxyglucose (inhibits hexokinase 2) further supports targeting macrophage metabolism in PDAC [119].

## 7. Future Directions and Conclusions

Altered metabolism or metabolic reprogramming fuels the growing energy demand of pancreatic tumor cells to support growth, proliferation, and survival. Thus, targeting metabolic intermediates and enzymes such as hexokinase, PFK-1, ME2, LDH-A, MUC-1, MUC-13, FASN, GOT1, GLS, GLUD1, BCAT2, FASN, ASNS, alanine transaminase, and transporters, including SLC1A4 and SLC38A2, might serve as a fruitful strategy to attenuate tumor growth and enhance therapeutic outcome. Small molecules or compounds targeting these enzymes have been reviewed by various studies [24,124,125,126,127,128] and are under investigation (Table 1). Additionally, targeting TAMs, PSC-stroma collaboration, epithelial-mesenchymal transition, monocyte-myofibroblast transdifferentiation, collagen-derived proline (proline metabolism), ROS (decrease or increase), stromal factors, autophagy, macropinocytosis, and factors involved in stemness might enhance therapeutic outcomes. The metabolic intermediates and enzymes involved in metabolic reprogramming should be targeted in PDAC mainly due to the dependency of tumor cells on the alternative source of energy (a typical feature of PDAC), which is fueled by metabolic re-programming involving these enzymes and is responsible for tumor growth in a harsh, hypoxic, and poorly vascularized microenvironment due to the stroma. Further, the focus should be to investigate drugs that can penetrate or disrupt the stroma and target tumor cells.

Focused research on molecular and metabolic targets, as discussed here, will advance our understanding of the molecular abnormalities in PDAC and will pave the way for the development of personalized treatment modalities. microRNAs (miRs) are involved in regulating pathways involved in cellular activity and metabolism. miRs are also involved in carcinoma deriving steps and may up- or down-regulate the tumor development and progression by regulating angiogenesis (miR-135a, miR-23), immune evasion (miR-146, miR-152), metastasis (miR-30a-3p, miR-210-3p, miR-885-3p, miR-9), reprogramming energy metabolism (miR-7, miR-1, miR-150), sustained proliferation (miR-27-3p, miR-545), and evasion from tumor suppressors (miR-19a) [135]. Since miRs expression levels get deregulated and vary during tumor progression, targeting and regulating miRs expression might be a potential therapeutic strategy for personalized therapy. Increased angiogenesis supports the tumor growth, regulating angiogenesis in PDAC might be beneficial. Inhibiting angiogenesis via targeting angiogenesis factors has been associated with increased aggressiveness of the tumor [136]; however, targeting angiogenesis with tumor suppressor miRs might be an interesting strategy [137]. Wang et al. [138] reported that let-7a, a tumor suppressor, regulates angiogenesis by concomitantly down-regulating transforming growth factor beta (TGFβ) receptor III (TGFBR3). Overexpression of let-7a inhibits tube formation and reduces the migration rate of HUVECs by directly targeting TGFBR3, resulting in a defective TGFβ signaling pathway in these cells, decreased angiogenesis and tumor size. The role of tumor suppressor miR-145 and miR-200 has also been discussed. Doublecortin-like kinase 1 (DCLK1), a putative marker of intestinal and pancreatic stem cells, regulates the pluripotency and expression of angiogenic factors via miRNA-dependent mechanisms in pancreatic cancer. Sureban et al. [139] reported that the downregulation of DCLK1 upregulates tumor suppressor miRs, including miR-145, miR-200, and let-7a, thus inhibiting tumor growth, metastasis, and angiogenesis.

As discussed, most of the studies targeting these factors are either conducted in cells or mouse models. Development of better preclinical models mimicking human disease to investigate the targeted and personalized therapies elucidating DNA repair abnormalities, targeting mesothelin, disrupting the hyaluronan, and blocking metabolomic enzymes are warranted to translate the lab findings into clinics. Tumor subtyping, genomic profiling, genomic analysis for mutations in DNA damage repair pathways (BRCA1, BRCA2, KRAS, HER2, PALB2, ATM; IMPaCt trial), next-generation DNA and RNA sequencing to investigate the changing genomic landscape, characterizing post-transcriptional epigenetic regulation through miRNA and lnRNA regulating tumor growth are other potential strategies which should be focused on to develop targeted and personalized medicine. Further, focus to investigate a panel of biomarkers with efficacy including hyaluronan expression, mesothelin expression, DNA repair gene mutations testing, CA19–9, CEA, CA125, ACIN1, TNFRSCF10C, and diagnostic miRNA panels for evidence-based clinical practice should be of interest in targeted and personalized therapy for PDAC [135,136,137,138,139]. Overall, well-designed precise clinical trials with a combinatorial approach of immunotherapy, targeting signaling pathways, and DNA damage repair based on preclinical results after a precise patient selection to formulate potential therapeutics are encouraged. Developing a specific biomarker-oriented drug, knowledge of tumor subtypes or heterogeneity, risk-based monitoring and assessment, endpoint selection, understanding the challenges while conducting the personalized medicine clinical trials, cost-effectiveness, designing statistical significance, and keeping the patient’s perspectives while designing a clinical trial are important factors to be considered while designing and conducting a personalized clinical trial [140].

## Figures and Tables

**Figure 1 ijms-21-08502-f001:**
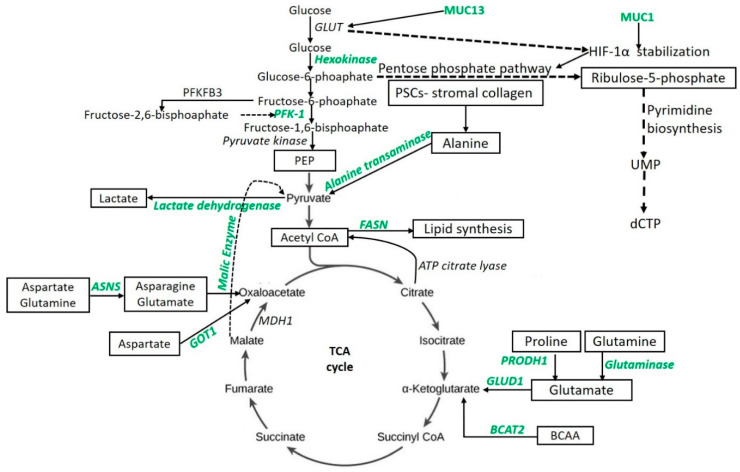
Schematic of targetable metabolic intermediates and enzymes (green) in pancreatic ductal adenocarcinoma. Aspartate aminotransferase (GOT1), asparagine synthetase (ASNS), branched-chain amino acid transaminase 2 (BCAT2), deoxycytidine triphosphate (dCTP), fatty acid synthase (FASN), glucose transporter (GLUT), glutamate dehydrogenase-1 (GLUD1), hypoxia-induced factor-1-alpha (HIF-1α), malate dehydrogenase-1 (MDH1), phosphofructokinase-1 (PFK-1), phosphoenolpyruvate (PEP), 6-phosphofructo-2-kinase 3 (PFKFB3), pancreatic stellate cells (PSCs), proline dehydrogenase (PRODH1), uridine monophosphate (UMP).

**Table 1 ijms-21-08502-t001:** Clinical trials targeting metabolomic enzymes in pancreatic cancer.

Phase	Trial Id	Target	Compound	Results/Status
I	NCT02071862	GLS	CB-839	Completed
III	NCT03504423	TCA cycle	FFX Versus CPI-613 with mFFX	Active
I	NCT01835041	TCA cycle	CPI-613 Versus CPI-613 with mFFX	Active
I/II	NCT01128296	autophagy	HCQ + gemcitabine	completed, encouraging results
I/II	NCT01506973	autophagy	HCQ + gemcitabine + abraxane	Active
II	NCT03601923	Poly ADP ribose polymerase	Niraparib	Active
II	NCT04409002	Poly ADP ribose polymerase	niraparib with dostarlimab (antibody attaching to protein called PD-1 on Tcells) and radiation therapy	Active
III	NCT03977272	PD-1	modified-FOLFIRINOX and Anti-PD-1 antibody in patients with metastatic pancreatic cancer.	Active
I	NCT03435289	PDH and α-KGDH	CPI-613 in combination with gemcitabine and nab-paclitaxel	unknown
I	NCT04181645	PD-1	SHR-1210 (anti-PD1)/Gemcitabine/Paclitaxel-albumin	Active
I	NCT03497819	mesothelin; tumor associated B cells	CARTmeso; CART19	Active
II/III	NCT03512756	Reactive oxygen species	SM-88 used with MPS (methoxsalen, phenytoin, sirolimus)	Active
II	NCT03509298	MUC1	Activated CIK and CD3-MUC1 Bispecific Antibody	Active
III	NCT03504423	Mitochondrial enzymes	FFX versus CPI-613 + mFFX	Active
I	NCT01839981	PDH	6,8-bis(benzylthio)octanoic acid	Completed, no results posted
II	NCT01273805	autophagy	Hydrooxychloroquine	Completed [129]
I	NCT01777477	autophagy	HCQ + gemcitabine	Completed [130]
I/II	NCT00096707	Hexokinase	2-DG alone or with docetaxel	No effects [131]
I/II	NCT00907166	PDH and α-KGDH	CPI-613 with gemcitabine	encouraging results [132]
II	NCT01167738	Mitochondrial complex I	metformin with PEXG	negative results [133]
II	NCT01210911	Mitochondrial complex II	metformin with Gemcitabine and erlotinib	negative results [134]

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
