# Peer review of "Targets (Metabolic Mediators) of Therapeutic Importance in Pancreatic Ductal Adenocarcinoma"

_ijms, 2020, doi:10.3390/ijms21228502_

Round 1
Reviewer 1 Report
This is an exhaustive review regarding metabolic mediators of therapeutic importance in pancreatic cancer: the review is well-written and the literature search is detailed.
Minor point: section Introduction, lines 46-47." Surgical resection of the tumor tissue is the only curative treatment available for locally advanced or metastatic PDAC". This is wrong, surgical resection is not indicated for locally advanced or metastatic tumors. The phrase should be reformulated.
Author Response
Comment: This is an exhaustive review regarding metabolic mediators of therapeutic importance in pancreatic cancer: the review is well-written, and the literature search is detailed.
Response: Thank you for your supportive comment.
Concern: section Introduction, lines 46-47." Surgical resection of the tumor tissue is the only curative treatment available for locally advanced or metastatic PDAC". This is wrong, surgical resection is not indicated for locally advanced or metastatic tumors. The phrase should be reformulated.
Response: Thank you for your comment and suggestion. We apologize for the mistake. We have reformulated the sentence in the revised manuscript as suggested (lines 43-46).
Reviewer 2 Report
This is an good review and balanced assessment of the status of metabolic mediators’ inhibitors and autophagy, macropinocytosis, lysosomal transport, recycling, amino acid transport, lipid transport, and the role of reactive oxygen species has also been discussed. The role of various pro-inflammatory cytokines and immune cells from an authority in the field. The article highlights important data that might have been overlooked when promulgating the clinical value of PDAC and related trials.
Few points need to be addressed.
This reviewer personally misses some novel insights reported about the role of the epigenetics in the regulation of tumor angiogenesis (i.e. by micro-RNAs; PMID: 31757094). Indeed, Let-7a inhibits tube formation and reduces the migration rate of HUVECs by directly targeting transforming growth factor beta (TGFβ) receptor III, resulting in a defective TGFβ signaling pathway in these cells. Moreover, in pancreatic cancer, doublecortin-like kinase 1 (DCLK1), a putative marker of intestinal and pancreatic stem cells, regulates the pluripotency and expression of angiogenic factors via miRNA-dependent mechanisms. The downregulation of DCLK1 upregulates several tumor suppressor miRNAs, including let-7a, thus inhibiting tumor growth, metastasis, and angiogenesis.
The authors should discuss these shreds of evidence: the underlying message here is that more precision and individualized approaches need to be tested in well-designed clinical trials – a challenge, but I would be interested in their perspective of how this might be done.
Author Response
Comment: This is a good review and balanced assessment of the status of metabolic mediators’ inhibitors and autophagy, macropinocytosis, lysosomal transport, recycling, amino acid transport, lipid transport, and the role of reactive oxygen species has also been discussed. The role of various pro-inflammatory cytokines and immune cells from an authority in the field. The article highlights important data that might have been overlooked when promulgating the clinical value of PDAC and related trials.
Response: Thank you for your comments.
Concern: This reviewer personally misses some novel insights reported about the role of the epigenetics in the regulation of tumor angiogenesis (i.e. by micro-RNAs; PMID: 31757094). Indeed, Let-7a inhibits tube formation and reduces the migration rate of HUVECs by directly targeting transforming growth factor beta (TGFβ) receptor III, resulting in a defective TGFβ signaling pathway in these cells. Moreover, in pancreatic cancer, doublecortin-like kinase 1 (DCLK1), a putative marker of intestinal and pancreatic stem cells, regulates the pluripotency and expression of angiogenic factors via miRNA-dependent mechanisms. The downregulation of DCLK1 upregulates several tumor suppressor miRNAs, including let-7a, thus inhibiting tumor growth, metastasis, and angiogenesis. The authors should discuss these shreds of evidence: the underlying message here is that more precision and individualized approaches need to be tested in well-designed clinical trials – a challenge, but I would be interested in their perspective of how this might be done.
Response: Thank you for your comments and suggestions. We have included a section on the role of epigenetics and the need of investigations and precise clinical trials for targeted and personalized therapy in PDAC (lines 648-689) in the revised manuscript.
Reviewer 3 Report
The Authors presented a very interesting review, summarizing the most importants therapeutic targest in PDAC pathology. Nevertheless, the Authors should add an important paragraph describing the surgery action an its relationship with chemo-therapy assets.
The sentence reported at line 46 of manoscript is completely wrong.
“Surgical resection of the tumor tissue is the only curative treatment available for locally advanced or metastatic PDAC, which can also be 47 associated with recurrent PDAC and poor quality of life”.
Probabily, looking in the metabolic situation of PDAC pathology, all surgical treatment might be considered an uneffective treatment. However, it could be necessari that the authors distinguished the three following typologies of PDAC pathients in front of the surgical option: Resectable, borderline resectable and unresectable. Nevertheless, Locally advanced PDAC (LA PDAC) is completely different PADC patient with respect to metastatic PDAC one (mPDAC).
Indeed, the description of surgical option (i.e. palliation too) must to be described in association with neo-adjuvant and adjuvant therapeutic regimens.
A new paragraph describing this part is (in our opinion), mandatory.
Author Response
Comment: The Authors presented a very interesting review, summarizing the most importants therapeutic targest in PDAC pathology. Nevertheless, the Authors should add an important paragraph describing the surgery action and its relationship with chemo-therapy assets.
Response: Thank you for your comment and suggestions. We have modified the manuscript accordingly.
Concern 1: The sentence reported at line 46 of manuscript is completely wrong. “Surgical resection of the tumor tissue is the only curative treatment available for locally advanced or metastatic PDAC, which can also be 47 associated with recurrent PDAC and poor quality of life”.
Response: Thank you for your comment. We apologize for the mistake. We have modified this sentence in the revised manuscript (lines 43-46).
Concern 2: Probabily, looking in the metabolic situation of PDAC pathology, all surgical treatment might be considered an uneffective treatment. However, it could be necessari that the authors distinguished the three following typologies of PDAC pathients in front of the surgical option: Resectable, borderline resectable and unresectable. Nevertheless, locally advanced PDAC (LA PDAC) is completely different PADC patient with respect to metastatic PDAC one (mPDAC). Indeed, the description of surgical option (i.e. palliation too) must be described in association with neo-adjuvant and adjuvant therapeutic regimens. A new paragraph describing this part is (in our opinion), mandatory.
Response: Thank you for your comments and suggestions. We have included the description of resectable, borderline resectable, and unresectable surgical categories of the PDAC with a focus on the need to precisely distinguishing the borderline resectable category in the introduction section (lines 43-63).
Round 2
Reviewer 3 Report
The authors correctly answered to reviewer's comments.
We haven't additional comments to the authors.